# The Driverless Triple-Wild-Type (BRAF, RAS, KIT) Cutaneous Melanoma: Whole Genome Sequencing Discoveries

**DOI:** 10.3390/cancers15061712

**Published:** 2023-03-10

**Authors:** Orsolya Pipek, Laura Vizkeleti, Viktória Doma, Donát Alpár, Csaba Bödör, Sarolta Kárpáti, Jozsef Timar

**Affiliations:** 1Department of Physics of Complex Systems, ELTE Eötvös Loránd University, 1053 Budapest, Hungary; 2Department of Pathology, Forensic and Insurance Medicine, Semmelweis University, 1085 Budapest, Hungary; 3Department of Bioinformatics, Semmelweis University, 1085 Budapest, Hungary; 4Department of Dermatology, Venerology and Dermatooncology, Semmelweis University, 1085 Budapest, Hungary; 5Department of Pathology and Experimental Cancer Research, Semmelweis University, 1085 Budapest, Hungary

**Keywords:** skin melanoma, BRAF, RAS, KIT wild type, whole genome sequencing

## Abstract

**Simple Summary:**

Malignant melanoma of the skin develops primarily, but not exclusively, on UV-exposed skin, where the most frequent histological forms are superficial spreading and nodular melanomas. In these tumors in the vaste majority of cases (~80%), the driver oncogenes are mutant BRAF, RAS and KIT. The genetic makeup of the triple-wild-type melanoma (BRAF, NRAS and NF1) has been known for some time, but those studies grouped together rare histopathological versions with common ones, as well as mucosal and even uveal ones. Here we used whole genome sequencing to genetically characterize the triple-wild-type melanoma (TWM), termed here as BRAF, RAS and KIT wild type, using the most common histological forms and excluding rare ones. All these tumors except one were UV-induced. In this driverless setting, we revealed rare oncogenic drivers known from melanoma or other cancer types and identified rare actionable tyrosine kinase mutations in NTRK1/3, RET and VEGFR1. Mutations of TWM identified genes involved in antitumor immunity, Ca^++^ and BMP signaling. Even with this comprehensive genomic approach, cases remained driverless in several instances, suggesting that unrecognized drivers are hiding among passenger mutations.

**Abstract:**

The genetic makeup of the triple-wild-type melanoma (BRAF, NRAS and NF1) has been known for some time, but those studies grouped together rare histopathological versions with common ones, as well as mucosal and even uveal ones. Here we used whole genome sequencing to genetically characterize the triple-wild-type melanoma (TWM), termed here as BRAF, RAS and KIT wild type (the most frequent oncogenic drivers of skin melanoma), using the most common histological forms and excluding rare ones. All these tumors except one were clearly induced by UV based on the mutational signature. The tumor mutational burden was low in TWM, except in the NF1 mutant forms, and a relatively high frequency of elevated LOH scores suggested frequent homologue recombination deficiency, but this was only confirmed by the mutation signature in one case. Furthermore, all these TWMs were microsatellite-stabile. In this driverless setting, we revealed rare oncogenic drivers known from melanoma or other cancer types and identified rare actionable tyrosine kinase mutations in NTRK1, RET and VEGFR1. Mutations of TWM identified genes involved in antitumor immunity (negative and positive predictors of immunotherapy), Ca^++^ and BMP signaling. The two regressed melanomas of this cohort shared a 17-gene mutation signature, containing genes involved in antitumor immunity and several cell surface receptors. Even with this comprehensive genomic approach, a few cases remained driverless, suggesting that unrecognized drivers are hiding among passenger mutations.

## 1. Introduction

Malignant tumors of the pigment cells have several forms: skin melanomas of UV-exposed skin (common superficial spreading—SSM; nodular melanoma—NM), acrolentiginous variants, and rare histological forms developed from blue nevi or deep penetrating nevi. It is important to note that the tumor mutational burden (TMB) of the UV-induced skin melanoma is among the highest of all cancers [1]. However, melanomas can develop in mucosal surfaces as well as in the uvea. It is now evident that the genetic backgrounds of these histological variants are very different [2]. Since the vast majority of melanomas are common skin melanomas, earlier studies focused on their genetic makeup. These studies identified three major drivers of UV-induced skin melanomas: the BRAF, NRAS and KIT mutant forms covering ~80% of these tumors [3,4]. It is also important that one of the most frequent mutations in skin melanomas affects the TERT promoter [2,4]. Other less frequent drivers of skin melanomas according to the The Cancer Genome Atlas (TCGA) database are RAC1, IDH1, MAP2K1, H- and K-RAS, ARID2, PPP6C, and DDX3X [5]. The suppressor gene palette is also heterogenous, involving P53, CDKN2A, PTEN, NF1, and RB1. It is of note that the tumor mutational burden is the highest in NF1-mutated tumors followed by NRAS mutants [6]. Molecular classification also established the co-occurring mutations and copy number variations (CNVs) in the three forms, identifying that CDKN2A, P53, CDK4 and IDH1 mutations can be partners of all of those drivers [2,5]. Chromosomal instability also characterizes skin melanomas resulting in the frequent amplification of CCND1 and MITF and loss of CDKN2A and PTEN [2,3,5]. Fusion genes in solid tumors may also occur, but in the case of melanoma these are very rare, affecting BRAF, ROS1, RET or NTRK [7]. The molecular classification of melanomas has practical consequences since the BRAF and KIT mutant forms can be treated by targeted therapies. Furthermore, the identification of rare fusion genes may open the door for the application of tumor-agnostic inhibitors [7].

In the literature, the triple-wild-type melanoma (TWM) refers to those where BRAF, NRAS and NF1 are not mutant [5,6,8,9]. However, in these genomic analyses, histologically heterogenous melanomas have been grouped together, frequently containing rare skin versions, mucosal and even uveal ones. Furthermore, the third largest molecular subgroup (5–15%), the KIT mutants, was also included in this category, in which this oncogenic driver is the major driver and a relevant therapeutic target [10]. Table 1 summarizes data of four major studies on the TWM. TCGA identified mutations of nine genes (including KIT, GNA11 and GNAQ) and CNVs of five genes in the TWM, but contained rare histological forms, mucosal ones and uveal ones [5]. An analysis of a larger TWM cohort, again containing KIT, GNA11 and GNAQ mutants, confirmed the TCGA results, except PDGFRA, but added FGFR3 and ERBB2 mutations to the list [9]. A much larger study on the TWM of the skin added mutations in another 14 genes to the list, but again also contained KIT, GNA11 and GNAQ mutants [8]. A more recent analysis of a small TWM cohort revealed mutations and CNVs of another three genes, and this cohort of skin melanomas was not only BRAF and RAS wild type, but KIT as well [11].

Here we used the triple-wild-type term for those melanomas, where the usual oncogenic drivers, BRAF, RAS isoforms and KIT are wild type, and we used a cohort of the most common histological variants of skin melanomas—superficial spreading (SSM) and nodular (NM)—to see what kind of driver mutations were present and whether there was any unique actionability regarding these tumors, since they may represent a significant proportion of skin melanomas (~20%).

## 2. Materials and Methods

### 2.1. Patients

A frozen tissue biobank of melanomas was established consisting of 35 common skin melanoma cases: the cohort contained primary tumors and skin or locoregional lymph node metastases. In each case peripheral blood (PBL) was also collected. At first BRAF, NRAS and KIT mutations were determined by Sanger sequencing as described [12,13]. Only triple-negative cases were analyzed further, the clinical characteristics of which can be seen in Table 2. The study was conducted in accordance with the Declaration of Helsinki and approved by the Ethical Committee of the Medical Research Council of Hungary (ETT-TUKEB14383).

### 2.2. DNA Extraction and Quality Control

Genomic DNA was purified from fresh-frozen tissue samples and matched blood controls using QIAamp DNA Mini and QIAamp DNA Blood Mini Kits (BioMarker Kft., Gödöllő, Budapest, Hungary) according to the protocol of manufacturer. Quantity and quality of DNA were checked by using NanoDrop ND-1000 spectrophotometer (Thermo Fisher Scientific, Inc., Waltham, MA, USA) and Qubit dsDNA HS Assay on a Qubit 1.0 fluorometer (Thermo Fisher Scientific, Inc., Waltham, MA, USA). DNA samples with 260/280 ratio ≥ 1.8 were used for further analysis. Fragment analysis of gDNA was carried out by 1% agarose gel electrophoresis.

### 2.3. Whole Genome Sequencing

Next-generation sequencing libraries were prepared from 1 μg input material using the TruSeq DNA PCR-Free HT Library Prep Kit (Illumina) with IDT for Illumina TruSeq UD Indexes (Integrated DNA Technologies, Coralville, IA, USA). Briefly, genomic DNA was sheared using a Covaris S220 focused-ultrasonicator, DNA fragments were cleaned, end-repaired, and 3′ A-tailed, followed by ligation of the sequencing adapters. After quality control and quantification using the KAPA Library Quantification Kit Illumina^®^ Platforms (KAPA Biosystems, Wilmington, MA, USA), individual libraries were diluted, equimolarly pooled, and sequenced on an Illumina NovaSeq 6000 instrument using the Xp workflow with S4 flow cell and 150 bp paired-end chemistry. Library preparation and sequencing was performed in the Biomedical Sequencing Facility at CeMM—Research Center for Molecular Medicine of the Austrian Academy of Sciences (Vienna, Austria). The mutant detection sensitivity of this analysis was set at 3%.

### 2.4. Bioinformatic Analysis

Quality control of raw sequencing data was performed with the FastQC [14] and multiQC [15] software tools (GPL.3.0). Raw short reads were then aligned to the human reference genome (version hg38) with the bwamem [16] algorithm. Duplicate reads were marked with the SAMBLASTER [17] tool. Short somatic mutations in tumor-PBL sample pairs were detected using Mutect2 and further refined with the FilterMutectCalls GATK (Genome Analysis Toolkit) [18] tools. Short genomic variants were annotated with Ensembl Variant Effect Predictor (VEP) [19] using the ClinVar [20], dbSNP [21], COSMIC [22], 1000 genomes [23] and gnomAD [24] databases. COSMIC mutational signature decomposition was performed using an expectation maximization approach in R (version 4.2.1), while setting the list of initial mutational signatures to those frequently identified in melanoma cases (SBS1, SBS2, SBS3, SBS5, SBS7a, SBS7b, SBS7c, SBS7d, SBS13, SBS17a, SBS17b, SBS38, SBS40) [25]. CNV analysis was performed with the CNVkit software [26]. MSI status of the sequenced samples was determined with the MSIsensor2 tool [27].

## 3. Results

Whole genome sequencing revealed mutations of 317 genes in this triple-wild-type melanoma (TWM) cohort and the tumor mutational burden (WGS-TMB) was determined. The connection between WGS-TMB and panel TMB is that the 199-mutation WES-TMB corresponds to the 10 m/Mb panel TMB value [28]. This evaluation indicated that two cases were characterized by high TMB (cases 6 and 7). This analysis also showed that the majority of these TWMs contained a very low number of pathogenic mutations in the range of 4–53 (Table 3). Case 2 is unique since only a few genes contained pathogenic mutations, but contained 247 CNVs (Appendix A). mutational pattern analysis of this case identified an APOBEC signature, which is a clear indication of chromothripsis (Appendix A). The COSMIC mutational signature analysis revealed that these TWMs were predominantly induced by UV, except case 2 (Appendix A). Furthermore, we saw an association between the level of chronic sun damage (CSD) and TMB. The non-CSD case (2) had a very low TMB, while the two high-TMB cases were high-CSD (cases 6 and 7). On the other hand, the CNV frequencies of the TWMs were in the range of 2–247 and there was no connection evident between TMB and CNV burdens. Although in three cases a high LOH score was detected as a sign of homologue recombination deficiency (HRD) [29], mutational signature analysis confirmed the HRD-type only in case 5 (Appendix A). Furthermore, none of these TWMs were microsatellite-instability (MSI)-positive (Appendix A and Table 3).

Looking for recurrent mutations (>2 of the cases), we identified 19 genes (Table 4, Appendix A), but only one qualified for the role of driver oncogene, TERT (promoter) presents in all but one cases, and only CTNND2 and ZFPM2 qualified for the role of oncosuppressor. As compared to the TCGA database, BMPER, BRINP2, CTNND2, LAMB4, MUC4, MROH2B, POM121L12 and ZFPM2 mutations were over-represented in our TWM cohort (Table 4).

Next, we attempted to construct the driver patterns of the TWM, individually analyzing the pathogenic mutations and CNVs in each case (Table 5 and Appendix A). We found that the common melanoma suppressor gene alterations CDKN2A, CTNNB1, NF1, PTEN and TP53 could be identified in five of seven cases. It is of note that the NF1 mutant cases were found to be characterized by a high TMB. Looking for other potential oncosuppressor gene mutations, we found CTNND2 [30], HNF1A [31], TACC2 [32], TPTE2 [33] and ZFPM2 [34] mutations, leaving only one case without a probable oncosuppressor mutation (case 1).

Most of the mutations found in either driver genes or potential drug target genes were likely induced by UV-light exposure. This was determined based on the sequence context and the specific base substitution of each mutation and whether these were prominently present in any of the UV-related COSMIC mutational signatures (SBS7a, SBS7b, SBS7c, SBS7d, SBS38 and DBS1). The only notable exception was the PTEN gene affected in case 4 by a likely non-UV-related mutation.

Concerning oncogenic drivers, the TERT promoter mutation was present in all cases except case 2, and KRAS and NRAS amplifications were detected in two cases. In the TWM cases, except case 1, ARID3A [35], ASXL1 [36], ASXL2 [36], FMN2 [37], FAM83B [38] and ZFHX4 [39] mutations were also identified as potential oncogenic drivers, known from other tumor types. Furthermore, PARP4 and PARP14 mutations were revealed in some cases. Case 2 again seemed to be unique, since it barely contained gene mutations and only one possible driver (FMN2), but had a high CNV burden, with additional LOHs of several oncosuppressors, and contained amplifications of three drivers (KRAS, NRAS and NTRK1).

Concerning actionability, potential drug targets could only be found in four cases of TWMs (case 1: IDH1 and NTRK1; case 2: NTRK1; case 6: NTRK3 and VEGFR1; case 7: RET and VEGFR1), leaving three cases without therapy options other than immunotherapy. It is of note that NTRK1, RET and VEGFR1 mutations have all been characterized by ClinVar as variants of unknown significance (Appendix A). It is of note that in these three cases AHNAK2, MUC4, MUC16, and MUC17 mutations were found, which were previously reported to be involved in antitumoral immune mechanisms of melanoma [40,41,42,43]. Furthermore, CSMD1 mutations occurred in three of the seven cases, which in melanoma produce neoantigens mimicking bacteria (Burkholderia pseudomallei), a positive predictor for anti-CTL4 therapy [44]. Moreover, TTN mutations were also prevalent in the TWM, reported to be involved in antitumoral immune mechanisms [45]. It is also of note that the Ca-signaling pathway was involved in cases 2, 6 and 7, since mutations of RYR1, RYR2 and TRVP6 and amplification of RYR2 were detected. In four cases (cases 2, 3, 6 and 7), genetic alterations of BMP signaling were detected (BMPER and BRINP2 mutations). It may be important that the majority of these driver-like genes were interferone, which are regulated according to the Interferome database [46] (Table 5).

Melanoma (similar to other cancers) is a clonally heterogenous tumor, and using the variant allele frequency counts, we categorized the drivers and suppressors as clonal ones (reaching 50%, equal to 100% of the tumor cells if heterozygous), polyclonal ones (20–45%) or subclonal ones (<10%, present in <20% of tumor cells) [47]. This analysis demonstrated that in case 1, the mutant TERT promoter was a clonal driver, while there were two polyclonal drivers (NTRK1 and PARP14), and the others were subclonal. In case 2, the putative driver FMN2 was polyclonal. In case 3, the potential oncogenic drivers ASXL1 and TERT were subclonal, qualifying a driverless case. In case 4, the PTEN suppressor was clonal and TERT was also clonal, and the other potential oncogenic drivers ASXL2 and ZFHX4 were all polyclonal. In case 5, there was a clonal suppressor, TP53, and a clonal TERT driver, while the other potential oncogenic driver FAM83B was polyclonal. In case 6, there were four clonal suppressors, NF1, TP53, TACC2 and ZFPM2, and TERT was polyclonal but VEGFR1 was a clonal oncogenic driver, the latter also being a possible reliable drug target. In case 7, TERT was a clonal driver but the others were polyclonal. It is of note that in case 6, the Ca-signaling gene mutations were all clonal, as well as BRINP2, further suggesting a selection advantage for tumor cells with such mutation types. The observed mutations of genes involved in antitumor immune responses were mostly polyclonal, but mutations in CSMD1, MUC16 and TTN occurred as clonal alterations, suggesting again a clonal advantage for those tumor cells (Table 5 and Appendix A).

It is of note that our cohort contained TWMs of different developmental stages from the primary tumor to lymphatic metastasis. Neither driver patterns nor other genetic characteristics corresponded to those stages, except the TERT promoter mutation, which was clonal in primary tumors, local recidive or skin metastases, but became polyclonal or subclonal in lymphatic metastases (cases 3 and 6) (Table 5).

Lastly, our cohort contained two cases where the skin primary tumor was massively regressed (cases 3 and 6). Genetically, the two cases seemed to be different, since case 3 was characterized by very low tumor mutation and CNV rates, unlike case 6, although both were UV-induced. The two tumors had a common 18-gene mutation pattern overlap including oncogenic TERT and TACC2 oncosuppressor and BMP signaling (Table 5, Appendix A). Concerning gene mutations involved in antitumoral immunity, the two regressed cases shared AHNAK2, MUC4, and MUC16 gene mutations. Furthermore, there were several transmembrane receptor mutations shared by the two regressed cases: DSCAM, IGSF21, GHR, GRIA2 and RP1, offering common surface neoantigens for antitumoral immune reactions (Appendix A).

## 4. Discussion

The predominant histological forms of cutaneous malignant melanomas are superficial spreading and nodular ones, but the rest are pathologically and genetically very heterogenous [2,3,4]. In the majority of the most common histological forms, cutaneous malignant melanoma is a genetically well-defined tumor with clear driver and suppressor profiles dominated by BRAF and CDKN2A mutations. However, when the three major drivers, BRAF, RAS and KIT, are wild type, the remaining tumors, called here triple-wild-type melanomas (TWMs), are genetically very heterogenous and very different from the rare histological ones. It is of note that the mutational signature analysis confirmed UV exposure as the etiological factor in TWMs, except in one case. One interesting finding of our study is that when KIT mutant tumors were excluded, alterations of GNA11, GNAQ and KDR were not detected, although they were present in almost all previous TWM analyses [5,8,9]. A plausible explanation for this is that we analyzed here the common histological forms of cutaneous melanomas, SSM and NM. The NF1 mutants are characterized by a higher TMB [6,8], similar to our cases, but the other TWM cases were of very low TMB. The two high-TMB cases (cases 6 and 7 above the 175 m/Mb exome) are qualified for immune checkpoint inhibitor therapy. It is also of note that there was a clear connection between the level of CSD and TMB: the lowest TMB case was without CSD, while high-CSD cases were also of high TMB. On the other hand, a significant proportion of these TWMs were characterized by chromosomal instability, resulting in a relatively high CNV burden. Interestingly, gene mutations of the major homologue recombination repair genes are rare in melanoma [5,8] and it seems that the functional, epigenetic inactivation may be more frequent [8]. Although some of the TWMs had a high LOH score, the mutational signature analysis only revealed HRD in one case.

Reconstruction of the driver profile of the TWMs demonstrated that the TERT promoter mutation was a common driver in these tumors, in addition to KRAS and NRAS amplifications and the IDH1 mutation. However, in these TWMs, exotic drivers, ARID, ASXL, FMN2, FAM83B or ZFHX4 mutations may join the driver profile, although none of them were clonal, except ARID3A. A similar construction of the suppressor profile of TWMs resulted in a much more usual picture containing CDKN2A, CTNNB1, PTEN, TP53 LOHs as well as clonal NF1 and PTEN mutations. Furthermore, unusual suppressor mutations of CTNND2, HNF1A, TPTE2 and clonal alterations of TACC2 or ZFPM2 were observed. Wherever we referred to these as potential exotic drivers or oncosuppressors, we referred to the literature data. Experimental verification in human melanomas of the function of those genes is out of the scope of this paper but could be the basis of future studies. Interestingly, even with such a comprehensive genomic profiling, still there were TWM cases where a clear driver or suppressor gene alteration were not identified, suggesting that those genes may hide among the so called “passenger” mutations.

Concerning actionability, TWMs are difficult tumors, although in four of seven cases kinase gene alterations were identified in NTRK1, RET and VEGFR1, although all were of variants of unknown significance types. It is of note that the VEGFR1 mutation was clonal in one case, suggesting a possible driver role.

It was an interesting observation that two signaling pathways were identified in our TWM cohort: Ca^++^ signaling (RYR and TVPM6 mutations) and BMP (BMPER and BRINP2 mutations), which have recently suggested oncogenic functions in other cancer types [48,49]. RYR mutations are relatively common in melanomas, but their occurrence in low-TMB tumors may upgrade the significance of such mutations as possible novel drivers. It was further suggested by this observation that all the genes of Ca^++^ signaling were clonal in one TWM case.

The driverless nature of TWMs is a well-known phenomenon [6,8], and this is specifically true for those without KIT mutations. Nearly half of our TWMs were found to be lacking actionable gene mutations. In those cases, the remaining therapeutic option is only immune checkpoint inhibition. However, mutations in genes that are involved in immunoresistance of cancers such as ASXL [50], MUC4, MUC16 or MUC17 [40,41,42,43] may negatively affect the efficacy. On the other hand, positive predictors of immune checkpoint inhibitors such as a high TMB and mutations of CSMD1 or TTN may identify cases specifically prone to such therapies. It is of note that those three driverless cases were lacking CSMD1 mutations, but one contained a clonal TTN. Concerning antitumoral immunity, two TWM cases were regressed melanomas caused by natural antitumoral reactions. These two cases shared a common 18-gene mutation pattern, which included TERT, AHNAK2, MUC4 and MUC17. It is of note that AHNAK2 was recently reported to be a positive predictor of immunotherapy in lung cancer [41]. Furthermore, these two cases shared a common five-member mutant cell surface receptor signature as well (DSCAM, IGSF21, GHR, GRINA2 and RP1), suggesting that they may also have been involved in the massive antitumoral response to the primary tumor.

## 5. Conclusions

Although the term TWM has been known for some time, it was categorized as BRAF, NRAS and NF1 wild type, which contained the KIT mutant as well as the rare RAS mutant forms (KRAS and HRAS). Here, we used the TWM term for cutaneous melanomas where all the major oncogenic drivers are wild type: BRAF, RAS and KIT. Furthermore, our patient cohort contained common melanoma histotypes and excluded rare ones. We showed that the comprehensive whole genome sequencing of TWMs is a very useful approach to better characterize this type of cutaneous melanoma, to reveal novel drivers and suppressors, and to find therapy options for those patients. WGS or WES would have to be a clear choice over panel sequencing in TWM cases, which comprise 15–20% of these tumors, a clinically significant patient population.

## Figures and Tables

**Table 1 cancers-15-01712-t001:** Gene alteration patterns of triple-wild-type melanoma in the literature.

REF [5]		REF [9]	REF [8]	REF [11]	
mutation	CNV	mutation	mutation	mutation	CNV
CDKN2A	CCND1	CTNNB1	AP2B1	CCND1	CCND1
CTNNB1	CDK4	ERBB2	ARID2	CDKN2A	CDKN2A
EZH2	KDR	FGFR3	CDKN2A	FGFR3	
GNA11	MDM2	GNA11	CTNNB1	RAD51	
GNAQ	PDGFRA	GNAQ	FBXW7	RAF1	
IDH1		KIT	FRY	RB1	RB1
KIT		KDR	GNA11		PTEN
PTEN			GNAQ		IGFR1
TP53			IDH1		
			KIT		
			LZTR1		
			MAP1B		
			MAP2K1		
			MLH1		
			NPC1		
			RQCD1		
			SF3B1		
			SV2C		
			SLC39A10		
			TP53		
			ZMYND8		

CNV = copy number variation; triple-wild-type melanoma: BRAF, NRAS, NF1 wild type.

**Table 2 cancers-15-01712-t002:** Clinical data of the cutaneous triple-wild-type melanoma cohort.

Case No.	Gender	Age (Year)	Tumor Type	Primary Localisation	BR (mm)	Histological Type	CSD
1.	male	34	skin metastasis	lower extremity	3.85	NM	low
2.	female	51	skin metastasis	glabrous plantar skin	4.4	SSM	none
3.	female	44	LND metastasis	pubic	-	regressed	low
4.	female	89	local recidive	back	8.5	NM	low
5.	male	68	skin primary	chest	4.2	SSM	low
6.	male	58	LND metastasis	scapular skin	1.1	regressed	high
7.	female	79	skin primary	auricule	6.7	NM	high

BR = Breslow thickness; CSD = chronic sun damage; LND = lymph node; NM = nodular melanoma; SSM = superficial spreading melanoma.

**Table 3 cancers-15-01712-t003:** Gene alteration burden and mutation signatures of triple-wild-type melanoma.

Case	1.	2.	3.	4.	5.	6.	7.
total N of mutations	85,846	2576	28,148	41,635	35,410	293,162	539,258
pathogenic mutation N/exome	53	4	27	24	27	189	315
WGS-TMB/Mb	25.8	0.8	8.5	12.4	10.6	88.8	163.3
UV signature	predominant	minor	predominant	predominant	predominant	predominant	predominant
MS	MSS	MSS	MSS	MSS	MSS	MSS	MSS
total N of CNV	3	247	2	71	152	99	73
CNG							
A	0	39	1	0	6	11	4
TRS	2	37	1	58	56	21	55
CNL							
HL	1	1	0	0	0	0	0
LOH	0	170	0	13	90	67	14
LOH%	0	53.6	0	4.1	28.4	21.1	4.4
HRD signature	none	none	none	none	minor	none	none

A = amplification; CNG = copy number gain; CNL = copy number loss; CNV = copy number variation; HL = homozygous loss; HRD = homologous recombination deficiency; LOH = loss of heterozygosity; Mb = megabase DNA; MS = microsatellite; MSS = microsatellite stability; TMB = tumor mutational burden; TRS = trisomy; WGS = whole genome sequencing.

**Table 4 cancers-15-01712-t004:** Top mutated genes in triple-wild-type melanoma.

Gene Name	Symbol	Incidence (%)	TCGA (%)
ankyrin 3	ANK3	4/7 (57)	38
BMP binding endothelial regulator	BMPER	3/7 (43)	12
BMP/retinoic acid inducible neural-specific 2	BRINP2	3/7 (43)	13
complement 6	C6	3/7 (43)	24
catenin delta 2	CTNND2	3/7 (43)	14
CUB and Sushi multiple domain 1	CSMD1	3/7 (43)	40
dynein axonemal heavy chain 5	DNAH5	3/7 (43)	56
laminin B4	LAMB4	3/7 (43)	15
mucin 4	MUC4	4/7 (57)	18
Maestro heat-like repeat family member 2B	MROH2B	3/7 (43)	7
mucin 17	MUC17	3/7 (43)	31
Piccolo presynaptic cytomatrix protein	PCLO	4/7 (57)	49
POM121 transmembrane nucleoporin like 12	POM121L12	3/7 (43)	13
reelin	RELN	3/7 (43)	27
telomerase reverse transcriptase	TERT	6/7 (86)	77
titin	TTN	4/7 (57)	80
unc-1 homolog C3	UNC13C	3/7 (43)	30
Zinc finger homeobox protein 4	ZFHX4	4/7 (57)	33
Zinc finger protein, FOG family member 2	ZFPM2	3/7 (43)	18

Genes with more than a two-fold increase in incidence in our dataset compared to that of the TCGA-SKCM samples are highlighted with grey.

**Table 5 cancers-15-01712-t005:** Individual driver patterns of triple-wild-type melanoma.

Gene Alteration		Case No	1.	2.	3.	4.	5.	6.	7.
**drivers**	TCGA%	IRG							
CDKN2A	38			**LOHwt**		**LOHwt**	**LOHwt**	**LOHwt**	**LOHwt**
CTNNB1	7			**LOHwt**					
NF1	18	**+**		**LOHwt**				**C**	**P**
PTEN	16	**+**		**LOHwt**		**C**			
TP53	16	**+**					**LOH/C**	**C**	
IDH1	6	**+**	**S**						
KRAS	3	**+**		**Awt**			**Awt**		
NRAS	29			**Awt**					
TERT promoter	77		**C**		**S**	**C**	**C**	**P**	**C**
**actionable drivers**									
NTRK1	10	**+**	**P**	**Awt**					
NTRK3	12	**+**						**Awt**	
RET	8	**+**							**P**
VEGFR1	12							**C**	**P**
**potential drivers**									
CTNND2	14	**+**				**P**	**LOHwt**	**P**	**P**
HNF1A	5					**P**			
TACC2	25	**+**			**S**			**C**	
TPTE2	10				**S**				
ZFPM2	18	**+**				**P**		**C**	**P**
ARID3A	2.7	**+**						**C**	
ASXL1	5	**+**			**S**				**P**
ASXL2	7	**+**				**P**			**P**
FAM83B	20						**P**		
FMN2	24			**P**					
PARP4	9	**+**							**C**
PARP14	9	**+**	**P**						**P**
WNT7A	5								**C**
ZFHX4	33	**+**	**S**			**P**		**P**	**P**
**immunity**									
AHNAK2	24	**+**			**S**			**S**	
CSMD1	40		**P**					**C**	**P**
MUC4	18	**+**			**P**		**S**	**S**	**S**
MUC16	74				**S**			**C**	**P**
MUC17	31				**S**	**P**			**P**
TTN	80	**+**	**P**	**LOHwt**		**C**		**C**	**C**
**Ca signaling**									
RYR1	33							**C**	**P**
RYR2	32			**Awt**				**C**	**P**
TRVP6	8							**C**	
**BMP signaling**									
BMPER	12	**+**		**Awt**	**S**			**P**	**P**
BRINP2	13	**+**		**Awt**	**S**			**C**	**P**

A = amplification; C = clonal; HRD = homologous recombination deficiency; IRG = IFN-regulated gene; LOH = loss of heterozygosity; P = polyclonal; S = subclonal; wt = wild type. Blue = oncosuppressor; red = oncogenic driver; green = immunity-associated gene alterations; yellow = Ca signaling.

## Data Availability

A processed list of somatic mutations and CNV results are available at https://github.com/pipekorsi/TWM, accessed on 23 January 2023.

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
