# Peer review of "The Driverless Triple-Wild-Type (BRAF, RAS, KIT) Cutaneous Melanoma: Whole Genome Sequencing Discoveries"

_cancers, 2023, doi:10.3390/cancers15061712_

Round 1
Reviewer 1 Report
The authors used whole genome sequencing to genetically characterize a small cohort of triple wild type melanoma (TWM), without the most frequent oncogenic drivers of skin melanoma as BRAF, RAS and KIT using common histological forms as nodular melanoma (NM) and superficial spreading Melanoma (SSM).
1) Table 3 Gene alteration burden and mutation signatures of triple wild-type melanoma
Does not provide a clear breakdown of frequency of CNV. A and TRS are referred to the "Copy number gain", and HL and LOH are referred to the "CNV loss of heterozygosity". Adding the two subgroups the table will be clearer
2) The sum of the values in the CNV column for sample 2 is not 257 as reported but probably 247.
3) Discussion pg. 7/12 “ The NF1 mutants are characterized by a higher TMB [6,8], similar to our cases but the other TWMs were of very low TMB-values and none qualifies for immunotherapy based on this genetic marker”.
Why in the discussion the authors say the following sentence?
Several studies have evaluated the relationship between tumor mutational burden (TMB) and outcomes of immune checkpoint inhibitors. In the phase II KEYNOTE-158 study of pembrolizumab monotherapy for previously treated recurrent or metastatic cancer, High TMB can improved objective response rate (ORR). TMB ≥175 mutations/exome or more than 10 mutations/Mb is associated with clinically meaningful improvement in the efficacy of pembrolizumab monotherapy and improved outcomes for pembrolizumab versus chemotherapy across a wide range of previously treated advanced solid tumor types.
Interesting values for cases 6 and 7 with regard to Pathogenic mutation N/exome and for WGS TMB/Mb are given in Table 3.
Author Response
We appreciate of the constructive critique of the reviewer.
- Table 3 is corrected according to the suggestion: CNG, CNL
- Sample 2. CNV sum was corrected for 247
- TMB of the two cases (6,7) now categorized as high TMB in Abstract, in Results and in the Discussion qualified for immunotherapy.
Reviewer 2 Report
In this study, Tímár J and his/her colleagues performed whole genome sequencing for 7 melanoma samples with common characteristics of BRAF, RAS, and KIT wild type, followed by bioinformatic analysis to pinpoint their potential drivers for oncogenesis. A series of findings have been documented in this current version of the manuscript, including the illustration of several potential actionable tyrosine kinase mutations in NTRK1/3, RET, and VEGFR1. Several questions need to be addressed before the consideration of possible publication in Cancers.
1. The authors could use the ClinVar database (https://www.ncbi.nlm.nih.gov/clinvar) to further analyze their sequencing results, and then generate a potential actionable tyrosine kinase mutation list of the BRAF, RAS, and KIT wild-type melanoma patients.
2. For the genes and mutations on the top of the list, the golden standard function assays (colony formation assay in vitro and xenograft assay in vivo) for defining oncogenes or tumor suppressors are required to be included in the revised version of the manuscript.
3. (Minor): typo error: Simple summary section: Line 3, “Waste” should be “Vast”.
Author Response
We appreciate the constructive critique of the reviewer.
- We have re-tested the NTRK1, RET and VEGFR1 mutations in ClinVar (as was done originally (Suppl. Fig1.) and these mutations are the VUS cathegory. This information is included into the Result and Discussion.
- This paper is a descriptive analysis of the gene mutations found in TWM. Wherever we refer to potential oncosuppressor or oncogenic driver function of a gene we refer to literature data. Experimental verification in human melanoma of the function of those genes is out of scope of this paper but could be the basis of further stdies.
- The typo was corrected in the Simple Summary
Reviewer 3 Report
This report describes the whole genome sequencing of some cases of TWMs (triple wild type melanoma concerning the BRAF, RAS and KIT genes) and a interesting discussion about the results obtained. The ultimate goal of this approach would be the designing of a better characterization of such melanomas.
The TWMs comprise 15-20% of melanoma tumors, so that authors consider that this is a clinically significant proportion. It is right that the finding of reveal novel drivers and suppressors would be interesting and useful information to find therapy options for those patients. It was known that these TWMs are genetically very heterogenous, and this study confirms that complexity, so that the approach does not offer any definitive simple conclusion about this heterogeneous type of melanoma. Mutations of 317 genes were revealed in the current (TWM) cohort. Obviously, this is difficult to classify
I am afraid that discoveries are not definitive. On the other hand, the identification of some tyrosine kinase mutations (in NTRK1/3, RET and VEGFR1) is valuable and promising. Therefore, under my view, the study, sequences, recompilation and bioinformatic study is interesting and it would be suitable for publication.
However, the following minor points should be addressed before definitive acceptance.
1) A higher effort for the clarification of the abbreviation should be made. In addition to the abbreviation list at the end of the manuscript. For example, the definition of melanoma, superficial spreading (SSM) or nodular melanoma (NM) should be introduced the first time that the abbreviation is used, not later. In this case, SSM and NM are used at page 2, just at the beginning of introduction, but the definition is introduced at page 3,
2) Periods and other punctuation marks should be checked throughout the manuscript, especially periods close to references. Some of them are missing,, other wrongly introduced between two words in a sentence and many others are located before the reference at the end of the sentence, but they would be after the reference.
3) Letter size at the abbreviation list and references should be modified according the template of the Journal.
4) Suppl. material could be ordered, at the current state is not very useful. Most of those unordered data are already incorporated to regular Tables.
Author Response
We appreciate the constructive critique of the reviewer.
- Abbrews have been explained in the text at the first appearence.
- Periods and other punctuations were corrected thoughout the text to follow Cancers style.
- Letter size was corrected for Abbreviations and References
- We would like to keep the Supplementary Fig1. to give a solid foundation for our detailed analysis in the Text and in Tables. This also give an opportunity for anybody to re-analyse our data.
Round 2
Reviewer 2 Report
As the authors have not completed my request for revisions to this manuscript (point #2), I cannot be more positive for the publication of this work in Cancers.
Author Response
We appreciate the reviewers request to individually verify the oncogenic potential of the suspected oncogenic and suppressor mutations. We obtained their putative function from the literature, reffered to it and included this information into the Discussion. However this manuscript does not attemp to perform the experiments which could well be the topic of several papers...
Please accept this situation.